# Dense and Aligned Captions (DAC) Promote Compositional Reasoning in VL Models

Sivan Doveh[1,2]    Assaf Arbelle[1]    Sivan Harary[1]    Paola Cascante-Bonilla[7]

Amit Alfassy[1,5]    Roei Herzig[1,3]    Donghyun Kim[4,6]    Raja Giryes[3]    Rogerio Feris[4]

Rameswar Panda[4]    Shimon Ullman[2]    Leonid Karlinsky[4]

[1]IBM Research, [2]Weizmann Institute of Science, [3]Tel-Aviv University,
[4]MIT-IBM Watson AI Lab, [5]Technion, [6]Korea University, [7]Rice University

## Abstract

Vision and Language (VL) models offer an effective method for aligning representation spaces of images and text, leading to numerous applications such as cross-modal retrieval, visual question answering, captioning, and more. However, the aligned image-text spaces learned by all the popular VL models are still suffering from the so-called 'object bias' - their representations behave as 'bags of nouns', mostly ignoring or downsizing the attributes, relations, and states of objects described/appearing in texts/images. Although some great attempts at fixing these 'compositional reasoning' issues were proposed in the recent literature, the problem is still far from being solved. In this paper, we uncover two factors limiting the VL models' compositional reasoning performance. These two factors are properties of the paired VL dataset used for finetuning and pre-training the VL model: (i) the caption quality, or in other words 'image-alignment', of the texts; and (ii) the 'density' of the captions in the sense of mentioning all the details appearing on the image. We propose a fine-tuning approach for automatically treating these factors leveraging a standard VL dataset (CC3M). Applied to CLIP, we demonstrate its significant compositional reasoning performance increase of up to $\sim 27\%$ over the base model, up to $\sim 20\%$ over the strongest baseline, and by $6.7\%$ on average.

## 1 Introduction

Recently, with major discoveries in generative language [1–5] and vision [6, 7] modeling, and Vision & Language (VL) alignment techniques [2, 8–12], we are getting closer than ever to attaining models capable of complete understanding and knowledge of the surrounding world, as well as being capable of acting w.r.t. this knowledge. The 'eyes' of these emerging systems are the VL models that have been shown to exhibit strong performance in a wide range of applications, such as recognition [8, 13], detection [14–16], segmentation [17–19], visual question-answering [9, 10], captioning [9, 10] and many more. However, despite these great advances, the VL models are still known to suffer from significant drawbacks in compositional reasoning - the ability to understand (and properly reflect in the aligned VL representations) the non-object notions appearing on the images and in the text captions, such as object attributes, states, and inter-object relations. This has been publicized and extensively studied in several recent works that also propose metrics for analyzing this phenomenon [20–22]. This drawback of VL models may lead to undesired biases and misconceptions when using them as the 'eyes' of SOTA Large Language Models (LLMs) [23] to build the pinnacle of today's AI systems - multi-modal conversational AI [11, 12]. For example, Figure 1a, illustrates

37th Conference on Neural Information Processing Systems (NeurIPS 2023).

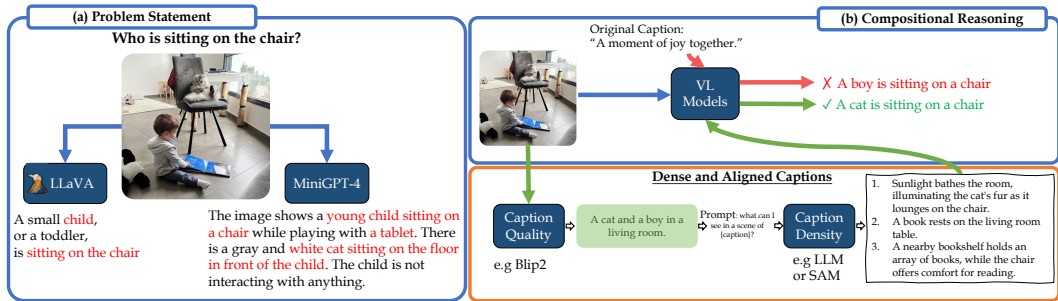

Figure 1: (a) Current VL models struggle with compositional aspects. In this case, both LLaVA and MiniGPT-4 which are extremely strong combinations of VL models with LLMs, misunderstand some basic spatial relations. (b) We are able to improve VL models' compositional reasoning with our proposed fine-tuning approach including improving caption quality and increasing caption density.

a failure mode of both Mini-GPT4 [11] and LLaVa [12], which successfully align one of the most advanced open-source LLMs (Vicuna [5]) with some of the best open-source VL models (BLIP2 [10] in Mini-GPT4 [11] and CLIP [8] in LLaVa [12]), demonstrating unprecedented capabilities for multi-modal chat, yet in some cases failing to recognize simple inter-object relations appearing on an image (Fig. 1a) despite successfully recognizing all the objects.

In this work, we offer a simple Dense and Aligned Captions (DAC) approach for enhancing the compositional reasoning abilities of VL models, such as the popular CLIP [8] model, without sacrificing their downstream transferability. On top of the previously proposed techniques of negative text augmentation [21, 24], we further study the effects of automatically improving the '*caption quality*' and '*caption density*' in a paired VL dataset on the compositional reasoning performance of VL models finetuned on the improved dataset (Fig. 1b). Surprisingly, we find that the low quality of the web-crawled captions (alt-texts, not designed to be image-descriptive), that fueled the enormous 400M [8] and 9B [25] paired image-text collections used to train the most widely used VL models [8–10], is one of the main reasons behind degraded compositional reasoning ability in these models. The main source of the low quality seems to be in the loose coupling many of the captions in these datasets have with their respective paired images. Examples include captions mentioning unrelated facts or human emotions, thus contributing a considerable amount of noise to the text-to-image representation spaces alignment learned by the VL model. Another aspect we found important for the degraded compositional reasoning performance seems to be the succinctness and partiality of the web-crawled captions. In the majority of cases, these captions do not describe the paired image in its entirety, but rather mention only a part of the image content, commonly missing object states, relations, and attributes appearing on the image. An intuitive effect of this is that many of the details visible to the model on the image side, are not represented on the text side and hence are being suppressed by the contrastive learning objectives driving the representation spaces alignment, leading to under-representation of those details in the resulting image and text embedding vectors.

Our proposed DAC approach automatically enhances the '*caption quality*' and '*caption density*' of the text captions in a paired VL dataset by using a combination of an image captioning technique [10], large language models [26, 27], and a foundation segmentation model [28]. In addition, we propose a finetuning approach to best leverage the enhanced VL dataset incorporating Multiple Instance Learning (MIL) losses and a variant of the previous SOTA negative text augmentation compositional reasoning improvement methods [21, 24]. Using our DAC, we are able to significantly boost the compositional reasoning performance of CLIP [8], measured on a variety of VL-checklist [20] and ARO [21] benchmarks, by up to $\sim 27\%$ in inter-object relations and by over $6.7\%$ over the highest baseline on average. Moreover, following our proposed VL models' improvements we observe almost no decrease in their linear probing accuracy - arguably the most important metric in light of the recent "linear alignment to LLM" based techniques [11, 12] successfully leveraging these VL models in multi-modal conversational AI systems closely mimicking GPT-4 [2].

To summarize, our contributions are as follows: (i) we propose a Dense and Aligned Captions (DAC) approach for enhancing a VL model's compositional reasoning performance via automatically enhancing the caption quality and density of any off-the-shelf VL dataset and applying the proposed fine-tuning technique utilizing Multiple Instance Learning and negative text augmentation; (ii) applying our proposed approach to the popular CLIP model and finetuning on CC3M enhanced

with our technique, we arrive at DAC-LLM and DAC-SAM, that demonstrate significantly higher compositional reasoning performance on large-scale VL-checklist [20] and ARO [21] benchmarks with up to 27% absolute improvement in inter-object relations over base CLIP and over 6.7% average improvement over the strongest baseline; (iii) we perform a detailed analysis and ablation of both our proposed DAC approaches, as well as of the importance of the caption quality and density factors for enhancing the compositional reasoning of VL models.

## 2 Related Work

**Vision-language (VL) Models.** There have been notable advances in large-scale vision-language (VL) models recently (*e.g.*, CLIP [8] and ALIGN [13]). These models are trained by aligning large-scale image and text pairs obtained from the web with contrastive alignment. Advanced image-text alignments method [9, 10, 13, 29, 30] have been introduced using cross-attention layers with supplementary unsupervised learning objectives including image-text matching, masked, and autoregressive language modeling or learn finer-level alignment and relations between image and text [31–35]. For example, BLIP [9] jointly combines multiple such objectives in a single multi-task training. CyClip [31] adds additional geometrical consistency losses in the image and text alignment. However, recent studies [20–22] reveal that VL models pre-trained on image-text pairs from the web lacks compositional understanding of image and do not understand structured concepts such as object attributes and relations. SLVC [24] and ARO [21] improve the compositional understanding of VL models by augmenting samples with negative texts. In a concurrent work, SGVL [36] leverages a dataset with structural scene graph supervision. Unlike these works, in this paper, we propose to enhance the caption quality (in terms of image alignment) and density (in terms of mentioning all details on the image) in a VL dataset without requiring any additional supervision, as well as propose a negatives-enhanced MIL finetuning method for leveraging the improved VL data to attain compositional reasoning performance significantly improving over the past approaches.

**Compositional Reasoning.** To achieve a compositional and structured understanding of visual scenes, it is essential to possess the capability to understand visual concepts, including the detection of individual entities and the ability to reason about their interactions and attributes. The ability of compositional reasoning has been successfully utilized in various computer vision applications including vision and language [37–40], scene graph generation [41–45], relational reasoning [46, 47], visual spatial reasoning [48], human-object interactions [49–51], action recognition [52–59], and even image & video generation from graphs [60–62]. Although these approaches have the potential to enhance compositional understanding, they heavily rely on dense and manually curated supervision, such as annotating the precise object locations and their relationships. However, collecting such annotations at scale can be extremely costly, leading to limited-size datasets or the use of synthetic data sources during training. In contrast, our work takes advantage of readily available foundation models, such as vision-language models, large-language models, and segmentation models, to enhance text data derived from noisy web image-text pairs. This augmentation process leads to improved captions, enabling the model to acquire the ability to engage in compositional reasoning regarding the image.

**Multiple Instance Learning.** Training with noisy labels in the form of 'bags of instances' has been widely researched and applied to many computer vision tasks [44, 63–73]. When dealing with MIL, multiple options have been explored to select a single candidate from the 'bag' such as maximum or random sampling with respect to some predefined measure. Other methods average over the whole bag using different averaging methods such as algebraic mean or MIL-NCE [64]. Recently FETA [74] introduced MIL losses adapted from MIL-NCE [64] to train a VL model with noisy labels generated from automatically from documents. In this work, we extend the prior work by incorporating negative examples generated from the MIL bag into the MIL loss. We demonstrate that this approach plays an important role in compositional reasoning improvement, particularly in VL, and in this capacity is superior to random, maximum, average, or NCE methods.

## 3 Method

In this section, we detail our DAC approach (Fig. 2) to enhancing the compositional reasoning performance of VL models (e.g., CLIP [8]). Our approach admits an arbitrary paired VL dataset (e.g., CC3M [75]) and applies automatic caption quality and caption density enhancements (Sec. 3.1 and Sec. 3.2). This enhanced dataset is then used to finetune the VL model by employing negative text augmentation (Sec. 3.3), Multiple Instance Learning (MIL) losses (Sec. 3.6) to cope with additional

noise introduced by the caption density enhancement step, additional losses (Sec. 3.5), and parameter efficient fine-tuning to reduce forgetting (Sec. 3.4).

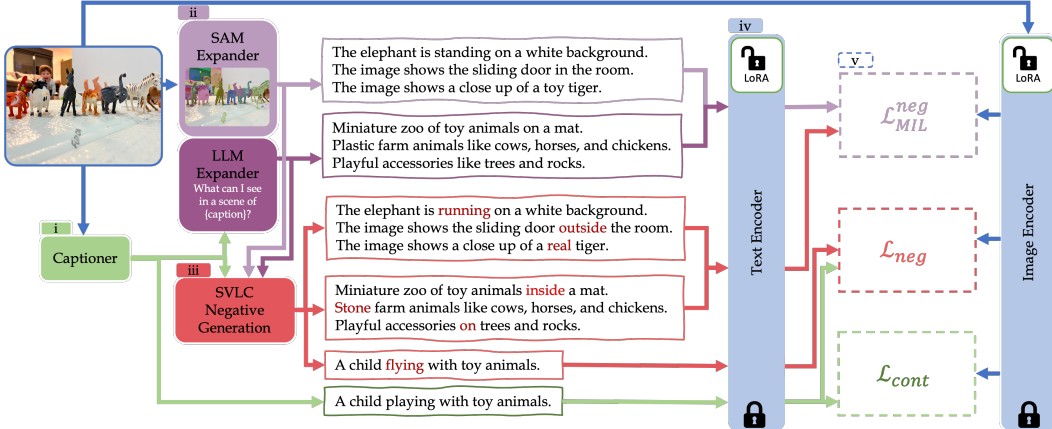

Figure 2: Detailed flow of our method: (i) The image is first captioned using the strong Captioner to create a high-quality caption. (ii) Then two methods for caption density enhancement (expansion) are applied: the "LLM-Expander" and the "SAM-Expander". Both create additional captions which relate to the image. (iii) the negative generator is applied to all captions, including the expanded ones. (iv) The image and all the captions are encoded using their respective encoders. (v) Finally, three losses are applied to the matching between the visual and textual embeddings.

## 3.1 Improving caption quality

One of the drawbacks of the Internet-collected VL datasets [25, 75] is the loose alignment the captions commonly have to their paired images. This likely stems from the heuristics commonly employed for this pairing, such as using co-location on the sources web-pages, alt-text, etc. Additional factors that impact the caption-to-image alignment, or in other words 'caption quality', are the ambiguous nature of the captions since different web-page authors refer to the images in different detail and include aspects like the human authors' intent, creativity, and many others. As noted in previous work [76], caption quality may negatively affect VL model transfer to downstream tasks. In this work, we analyze, for the first time, the direct impact that caption quality has on the compositional reasoning performance of VL models.

We employ a straightforward approach to enhancing the quality of the captions. To increase their alignment with their paired images, we use image captioning to generate text directly conditioned on the image. We use the BLIP2 [10] captioner, which in turn translates the image into 32 encoded tokens that are transferred to a language model (OPT [27]) which then outputs a sentence describing the content of the image. Sec. 5 includes an analysis of the effect of this proposed caption-to-image alignment. In particular, we show that: (i) the CLIP matching score statistics are significantly improved by replacing the original captions with BLIP captions (Fig. 3a), and qualitatively they are a better fit in many cases (Fig. 3b); (ii) even enhancing the caption quality alone has a positive impact on the compositional reasoning performance of the model; (iii) DAC full approach significantly outperforms BLIP2 [10] in terms of compositional reasoning (by over $6.4\%$ on VL-checklist [20] and over $50\%$ on ARO, Tab. 1), indicating caption quality enhancement alone is insufficient. Interestingly, a bonus property of the proposed caption enhancement method is that by using it one could potentially leverage any unlabeled image collection, without requiring any paired text data, for improving the compositional reasoning performance of a VL model. We believe that our observation, therefore, uncovers an interesting hidden potential of unlabeled image collections for VL training and may have a significant potential impact on the VL compositional reasoning performance scaling in future work, underlining the significance of our findings.

## 3.2 Improving caption density

Another important drawback of Internet-collected VL datasets is their 'partiality' or 'lack of density'. Intuitively, most of these captions are not intended by web page authors to be descriptive of the image but rather serve different purposes. Consequently, in most if not all of the cases, these captions do not tell the complete story of the image in terms of not mentioning every detail and not imparting

additional knowledge on related concepts from world knowledge. Below, we propose and explore two orthogonal methods for enhancing the density of the captions.

**Large Language Model (LLM) knowledge expansion.** One solution, is using either the original caption, the higher quality caption produced by an image captioner, or both, and use them as a prompt to a Large Language Model (LLM) effectively tapping on its world knowledge acquired from training on trillions of tokens. In our experiments we used the GPT-Neo-2.7B open LLM, instructing it to generate an expansion from a given caption via zero-shot (in context) prompts. The full prompts are long and provided in the Supplementary, their essence though can be summarized here as: "what should I expect to see in an image of {caption}?". We parsed the resulting text into a collection of individual sentences and employed them in our training via MIL losses (Sec. 3.6) in order to account for the caption noise expected with this kind of LLM expansion (Fig. 2). Indeed, the source caption is the only information provided to the LLM, and some facts are likely to be hallucinated and unrelated to the corresponding image. We provide an analysis of the effects of this way of increasing caption density through LLMs as well as of various related design choices in Sec. 5. Interestingly, LLM knowledge expansion alone has a mixed effect on the compositional reasoning performance, however, it is very effective in combination with other proposed techniques.

**Leveraging semantic over-segmentation.** An additional, more image-grounded way, to enhance caption density is to decompose the image into a set of segments with consistent language semantics and caption each segment individually or in connected segment groups. Consequently, we employ SAM [28] to produce a decomposition of an image into a collection of segments (seeded by a grid of points). We refer to this as 'semantic over-segmentation' as SAM applied in this way produces segments generally respecting semantic boundaries between objects, but commonly over-segments, highlighting object parts and other sub-object regions. We caption each segment using BLIP2 [10] resulting in a collection of captions for each image. As it is an over-segmentation, some of the resulting segment captions tend to be noisy (Fig. 2). Similar to the LLM-generated captions, we also employ these in our training through MIL (Sec. 3.6). The analysis of the segmentation-based expansion and effects of this way of caption density enhancement on the compositional reasoning performance, in general, is provided in Sec. 5.

### 3.3  Negative text augmentation

While in the VL datasets, e.g. CC3M [75] and LAION [25], each image-text pair constitutes a single sample, it was shown in recent works that it is possible to significantly enhance the effectiveness of these samples for teaching compositional reasoning to VL models via simple negative text augmentation [21, 24]. Using this approach, every image receives a set of captions, one caption correctly matching the image (the source caption), and one or several captions manipulated to be negative text samples via changing words within the caption, thus highlighting the importance of different compositional reasoning aspects (attributes, relations, actions, states) to the finetuned model. For our DAC approach we also employ this negative text augmentation strategy combining it with the DAC captions enhancement principles outlined in Sec. 3.1 and Sec. 3.2. Specifically, we explore applying this augmentation both to the BLIP2 captions (Sec. 3.5), as well as to the LLM-expanded and Segmentation-expanded captions by their combination with the MIL training losses (Sec. 3.6). Notably, our DAC approach attains significant improvements (e.g. over $12\%$ on VL-checklist [20]) in compositional reasoning performance over the recent methods that have proposed the negative text augmentation [21, 24], underlining the importance of the DAC factors outlined above. The effects of negative text augmentation are further explored in Sec. 5.

### 3.4  Avoiding forgetting with Parameter Efficient Fine-Tuning

In this work we explore fine-tuning a large-scale pre-trained VL model (e.g. CLIP [8], pre-trained on 400M image-text pairs) on a relatively smaller scale enhanced VL dataset (e.g. CC3M [75] with 3M image-text pairs). Intuitively, naively finetuning on the smaller dataset may incur 'forgetting' the powerful linear transfer capabilities of the base model (CLIP) that are of major significance for downstream applications, such as LLaVa [12] and Mini-GPT4 [11] enabling open, GPT4-like, multi-modal conversational AI. As this is undesirable, similar to other works [24, 77, 78], we employ parameter efficient finetuning, or more specifically LoRA [79], incorporated into all parameters of both vision and text encoders of the model to finetune the otherwise frozen model. As can be seen in Sec. 4.3, this strategy almost completely eliminates linear transfer performance loss only observing $\sim 1.5\%$ drop in the 5-shot setting.

### 3.5 Training Losses

A dual-encoder VLM (e.g., CLIP [8]) admitting text-image pair $(T, I)$ is comprised of: (i) an image encoder $e_I = \mathcal{E}_I(I)$; (ii) a text encoder $e_T = \mathcal{E}_T(T)$. The text-image similarity score is computed as:

$$\mathcal{S}(T, I) = \exp\left(\frac{\tau e_T^T e_I}{||e_T||^2 ||e_I||^2}\right), \tag{1}$$

where $\tau$ is a learned temperature parameter.

**Contrastive Loss.** As most contemporary VL models, we employ the contrastive CLIP-loss [8] as one of our losses for each batch $\mathcal{B} = \{(T_i, I_i)\}$ where texts $T_i$ of the text-image pairs are quality-enhanced as explained in Sec. 3.1.

$$\mathcal{L}_{cont} = \sum_i log\left(\frac{S(T_i, I_i)}{\sum_j S(T_i, I_j)}\right) + log\left(\frac{S(T_i, I_i)}{\sum_k S(T_k, I_i)}\right). \tag{2}$$

**Negatives Loss.** In addition to using negative text augmentation as part of our proposed MIL losses (Sec. 3.6), we employ the following 'negatives loss' applied to the quality-enhanced captions $T_i$:

$$\mathcal{L}_{neg} = \sum_i -log\left(\frac{\mathcal{S}(T_i, I_i)}{\mathcal{S}(T_i, I_i) + \mathcal{S}(T_i^{neg}, I_i)}\right). \tag{3}$$

where $T_i^{neg}$ is the negative text generated from $T_i$ according to the technique of [24].

### 3.6 Multiple Instance Learning (MIL)

For both kinds of caption density expansions, LLM-based and Segmentation-based, proposed in Sec. 3.2, we arrive with a bag of caption texts tentatively corresponding to the paired image. One approach to using those bags is random sampling or averaging in combination with the other losses. However, due to the inherent noise (e.g., $\sim 46\%$ for LLM-expansion measured on 100 random images) in both the density expansion methods (explained in Sec. 3.2), as can be seen from the ablations in Sec. 5, neither produces satisfactory compositional reasoning improvements. To better cope with this noise, we propose the MIL approach discussed next.

The basic MIL setting considers a set of $M$ captions $\{T_{i,m}\}_{m=0}^M$ such that at least one of these captions is a positive match to the paired image $I_i$. In our DAC approach, we extend the MIL-NCE [64] loss that was originally proposed for visual representation learning from uncurated videos. We adopt the MIL-NCE for our needs as follows:

$$\mathcal{L}_{MIL}^{base} = -\frac{1}{B}\sum_i^B \log \frac{\sum_m S(T_{i,m}, I_i)}{\sum_{j=1}^B \sum_m S(T_{j,m}, I_i)} \tag{4}$$

For any source of MIL bag (LLM / Segmentation), to combine negative augmentation, explained in Section 3.3, with MIL we modify the base-MIL loss $\mathcal{L}_{MIL}^{base}$ in Eq. 4 to incorporate the non-matching captions in the denominator of the equation:

$$\mathcal{L}_{MIL}^{neg} = -\frac{1}{B}\sum_i^B \log \frac{\sum_m S(T_{i,m}, I_i)}{\left(\sum_m S(T_{i,m}^{neg}, I_i)\right) + \left(\sum_{j=1}^B \sum_m S(T_{j,m}, I_i)\right)} \tag{5}$$

where $T_{i,m}^{neg}$ is the result of negative augmentation applied to the captions bag element $T_{i,m}$. Finally, the full finetuning loss of our proposed DAC approach can be written as:

$$\mathcal{L}_{DAC} = \mathcal{L}_{cont} + \mathcal{L}_{neg} + \mathcal{L}_{MIL}^{neg} \tag{6}$$

### 3.7 Implementation details

We used the popular ViT-B/32 OpenAI CLIP [8] original PyTorch implementation as our VL model in all the experiments. For caption quality enhancement (Sec. 3.1), we used the LAVIS implementation of BLIP2 with OPT 6.7B LLM. We used ViT-H SAM model [28] for the Segmentation-based density expansion (Sec. 3.2). Furthermore, we employed the GPT-NEO-2.7B LLM for the LLM-based density expansion (Sec. 3.2). During training, we set the batch size to 128 when training without density expansion (for ablations) and to 32 with density expansions. We used 6 v100 GPUs for 12 hours to train a model. We set the learning rate 5.0e-4, and use the AdamW optimizer over 5 epochs initializing with the CLIP weights. Our code is provided in the Supplementary, and it will be released upon acceptance together with our trained weights.

| | VL-Checklist | | | ARO | | | | Avg |
|---|---|---|---|---|---|---|---|---|
| | Object | Attribute | Relation | VG-R | VG-A | COCO | FLICKR | |
| CLIP[8] | 81.58 | 67.6 | 63.05 | 59.98 | 63.18 | 47.9 | 60.2 | 63.35 |
| BLIP2[10] | 84.14 | **80.12** | 70.72 | 41.16 | 71.25 | 13.57 | 13.72 | 53.27 |
| NegClip[21] | 81.35 | 72.236 | 63.525 | 81 | 71 | 86 | 91 | 78.01 |
| SVLC ([24]) | 85 | 71.97 | 68.95 | 80.61 | 73.03 | 84.73 | 91.7 | 79.42 |
| DAC-LLM (Ours) | 87.3 | 77.27 | 86.41 | **81.28** | **73.91** | **94.47** | **95.68** | **85.18** |
| DAC-SAM (Ours) | **88.5** | 75.829 | **89.75** | 77.16 | 70.5 | 91.22 | 93.88 | 83.83 |

Table 1: Experimental results on the VL-Checklist [20] and ARO [21] datasets to test different compositional reasoning aspects. The two bottom rows present two versions of full approach differing in the type of caption density source, i.e LLM-based and SAM-based. We see that both our methods outperform the baselines by a significant margin in almost all cases. While all methods other than BLIP2 [10] are CLIP-like dual-encoder architectures, BLIP2's heavier encoder-decoder architecture gives it some advantage on the VL-checklist evaluation. Still, we outperform it by a large margin. Our method improves the base CLIP [8] model by over 20% on average.

## 4 Experimental Results

### 4.1 Datasets

**Training:** We use the Conceptual Captions 3M (CC3M) dataset [75] to finetune CLIP for enhanced compositional reasoning performance using our proposed DAC approach. CC3M is a large set of 3 million image-text pairs automatically crawled from the internet. While the dataset includes captions for the images, we have found that enhancing the quality of the captions before finetuning, as described in Sec. 3.1, greatly improves compositional reasoning. For our full approach, we therefore discard the original captions from the dataset, which (as discussed in Sec. 3.1) uncovers yet another interesting potential of our method - the ability to leverage any unlabeled image collection.

**Evaluation:** We evaluate our method on two major benchmarks for VL compositional reasoning, VL-Checklist [20] and ARO [21], and also the Elevater [80] for image classification to verify linear transferability of the models (which is of significant importance for usability of the VL models in contemporary downstream applications such as LLaVa [12] and Mini-GPT4 [11]).

**VL-Checklist [20]** is a benchmark constructed from four datasets, namely Visual Genome [81], SWiG [82], VAW [83], and HAKE [84]. Each image is paired with two captions: a positive and a negative. Positive captions are part of original (paired image-text) datasets. Negative captions were constructed from positive captions by changing one word which changes the meaning of the sentence. The negatives are categorized as changes related to objects, attributes, or relations. Each category is further divided into sub-categories such as size, material, color, etc.

**ARO [21]** is a new compositional reasoning benchmark that includes positive or negative captions and additionally evaluates sensitivity to word order in the sentence. Word-order negative sentences are created by re-ordering the words, which changes the semantics of the sentence in aspects like attribution, relations, and the general meaning of the word order. ARO is constructed from images and captions from COCO [85], Flick30K [86], and Visual Genome [81].

**Elevater [80]** consists of 20 different datasets, including common classification datasets such as CIFAR100 [87], EuroSat [88], and others. We used the Linear-Probing classification protocol described in the Elevater Image Classification Toolkit [80] for evaluating the linear transferability of our resulting DAC models with improved compositional reasoning performance.

### 4.2 DAC for improving VL model's compositional reasoning performance

Our DAC approach includes caption quality and caption density enhancement steps, followed by a proposed fine-tuning technique that effectively combines several losses, including the proposed MIL loss $\mathcal{L}_{MIL}^{neg}$ explained in Sec. 3.6 to cope with the inherent noise of caption density expansion. Our main results are summarized in Table 1 and are compared to four strong baselines: (i) CLIP [8] which is also the source model that we improve; (ii) BLIP2 [10] which we also use for caption quality enhancement - to show our resulting models are significantly stronger in compositional reasoning, despite being based on a faster dual-encoder CLIP model (compared to 2-stage encoder-decoder BLIP2); and (iii) two SOTA methods on the compositional reasoning benchmarks, both based on negative text augmentation - NegClip [21] and SVLC [24]. Our DAC-LLM explores the use of LLMs, specifically GPT-NEO-2.7B, to expand the caption and generate additional information that

| | 5-shot | 10-shot | 20 shot | all-shot |
|---|---|---|---|---|
| CLIP [8] | 66.19% | 69.58% | 71.90% | 78.96% |
| DAC-LLM | 64.92% | 69.20% | 72.98% | 77.44% |
| DAC-SAM | 64.33% | 69.41% | 72.92% | 79.3% |

Table 2: Linear probing results on ELEVATER (20 datasets). As desired, we do not observe any significant degradation w.r.t. the baseline in terms of LP accuracy in any setting.

| LLM | Neg | MIL | Obj | Attr | Rel |
|---|---|---|---|---|---|
| ✓ | | MAX | 84.58% | 70.99% | 60.35% |
| ✓ | | AVG | 85.41% | 70.87% | 65.57% |
| ✓ | | MIL | 85.74% | 71.04% | 69.05% |
| ✓ | ✓ | MIL | **87.31%** | **77.22%** | **86.41%** |

Table 3: MIL ablations: we evaluate different strategies for MIL while also examining the effects of negative examples within the MIL loss, ($\mathcal{L}_{MIL}^{base}$ vs $\mathcal{L}_{MIL}^{neg}$).

is plausible to find in a scene described by the source (quality enhanced) caption. Our DAC-SAM leverages semantic over-segmentation [28] and converts image segments into short captions generated from them using [10]. Both these methods increase caption density and add information that was not necessarily available in the quality-enhanced caption. Both employ MIL (Sec. 3.6) combined with negative text augmentation (Sec. 3.3). As can be seen in Tab. 1, both our methods significantly improve on most compositional reasoning metrics with gains of up to 17% over the current SOTA. In section 5 we analyze the contribution of each individual component.

## 4.3 Preserving downstream linear transferability with DAC

We used the Linear-Probing classification protocol described in the Elevater Image Classification Toolkit [80] for evaluating the linear transferability to downstream tasks of our resulting DAC models (DAC-LLM and DAC-SAM) with improved compositional reasoning performance. This evaluation is of significant importance, as it demonstrates in a way the usability of the improved VL models in contemporary downstream applications, such as LLaVa [12] and Mini-GPT4 [11] multi-modal conversational agents, which use the linear alignment of VL model features to a strong LLM, e.g., Vicuna [5] in [11, 12].Table 2 summarizes the Elevater Linear Probing (LP) results. As prescribed by the standard Elevater LP protocol [80], we explore different amounts of training data for LP training, i.e. 5-shot, 10-shot, 20-shot, and all-shot. For all settings, we see that our DAC models are comparable or better than the baseline CLIP [8], only having 1.5% drop in 5-shot. These experiments show that our DAC VL models compositional reasoning enhancement approach did not degrade the representation power of the original CLIP model.

## 5 Ablation study

Below we perform an extensive ablation analyzing and highlighting the importance of caption quality and caption density factors towards improving compositional understanding of VL models. We find that each factor separately contributes to compositional reasoning performance, also improving over the baselines, while the best results are obtained by combining the factors. All experiments were conducted for the same number of epochs on the same dataset starting from CLIP pre-trained weights.

| | Quality | Neg | Density | MIL | VL-Checklist | | | ARO | | | | Avg |
|---|---|---|---|---|---|---|---|---|---|---|---|---|
| | | | | | Object | Attribute | Relation | VG-R | VG-A | COCO | FLICKR | |
| CLIP | | | | | 81.58% | 67.60% | 63.05% | 59.98% | 63.18% | 47.90% | 60.20% | 63.36% |
| A | | | | | 80.93% | 66.28% | 55.52% | 50.19% | 62.48% | 21.03% | 28.82% | 52.18% |
| | | | LLM | ✓ | 83.54% | 68.57% | 58.32% | 65.49% | 62.81% | 38.56% | 50.84% | 61.16% |
| | | ✓ | | | 85% | 71.97% | 68.95% | 80.61% | 73.03% | 84.73% | 91.70% | 79.43% |
| | | ✓ | LLM | ✓ | 86.08% | 72.15% | 70.48% | 74.80% | 67.80% | 84.10% | 88.30% | 77.67% |
| B | ✓ | | | | 84.41% | 71.03% | 63.94% | 67.26% | 64.76% | 29.43% | 40.60% | 60.20% |
| | ✓ | | LLM | | 85.00% | 70.97% | 70.27% | 66.44% | 65.05% | 77.64% | 85.30% | 74.38% |
| | ✓ | ✓ | LLM | | 87.26% | 74.25% | 81.97% | 72.38% | 69.07% | 81.22% | 80.52% | 78.09% |
| C | ✓ | ✓ | LLM | ✓ | 87.30% | **77.27%** | 86.41% | **81.28%** | **73.91%** | **94.47%** | **96.12%** | **85.24%** |
| | ✓ | ✓ | SAM | ✓ | **88.50%** | 75.83% | **89.75%** | 77.16% | 70.50% | 91.22% | 93.88% | 83.83% |

Table 4: Ablation Study: We split the table into three sections: A - examines the effect of LLM expansion without quality improvements. B - analyzes the caption quality and density improvement without the MIL loss, by randomly choosing one example from the bag as the caption. C - Our full method with the two density expansion variants, i.e LLM-based and SAM-based, and MIL.

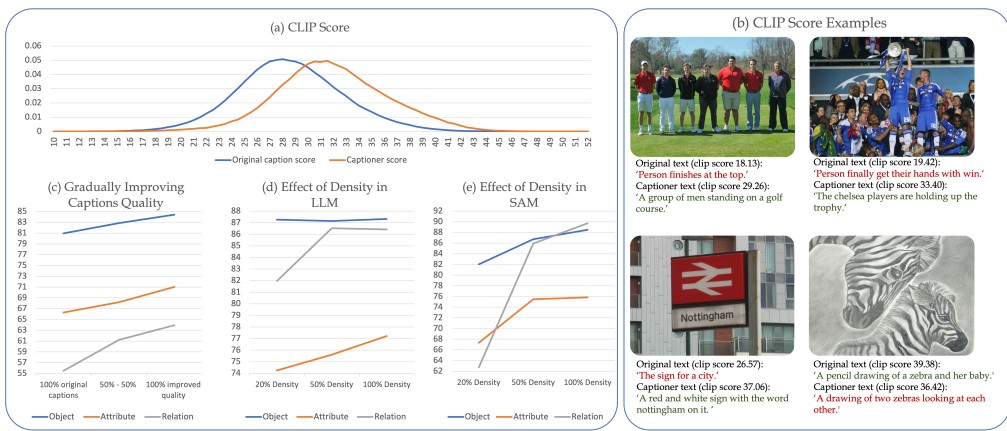

Figure 3: (a) The empirical probability density function of the CLIP score as a proxy of the caption quality. We see that the improved captions generally have higher scores than the original captions. (b) Examples of images with their original and generated captions. The captioner ([10]) can also make compositional reasoning mistakes (e.g. the zebra image), but our DAC can compensate through the proposed caption density expansion, as also evident from the quantitative evaluation advantage DAC has over [10] in Tab. 1. (c) Analysis of the VL-Checklist results with respect to the percent of captions replaced with our higher quality ones. We see a clear trend favoring quality captions. (d-e) The effects of improving caption density using LLM and SAM methods respectively, by increasing the percent of sampled captions, and thus the density, we see significant improvements.

**Caption Quality.** Here we validate the benefits of improving caption quality. First, Fig 3a compares the statistics of the CLIP score between captions and images before and after the quality improvement (Sec. 3.1). Clearly, improved captions have better matching statistics. Fig 3b shows some qualitative examples of caption improvement, in all cases original captions hardly describe the image content. Next, Figure 3c shows that gradually increasing the ratio of high-quality captions consistently improves the VL-Checklist metrics. To isolate the effect of quality improvement, we perform this experiment without the caption density enhancements. Finally, block A in Tab. 4 explores the contribution of DAC components without enhancing caption quality. We can see that all of them improve, yet the largest improvements are obtained with quality enhancement (blocks B and C). We have also conducted a caption quality user study. We asked 121 human subjects to review 110 random images from CC3M. For each image, we asked which of 3 choices best fits the image. A and B were captions, either produced by BLIP2 or originating from CC3M (A and B assignment was randomized). Option C was "neither caption fits the image". We found that 80.7% of responses favored the BLIP2 caption, 2.2% the original caption, and 17.1% was "neither". This indicates that BLIP2 captions are indeed better aligned with human perception than the alt-text collected from the Web in CC3M. Intuitively, such better alignment to humans, who are inherently good at compositional reasoning, likely leads to the significant compositional reasoning improvements observed when gradually increasing the percent of the higher quality (BLIP2) captions in the fine-tuning data (Fig. 3c).

**Caption Density.** Next, we validate the contribution of caption density expansion - increasing the number of captions from different sources (LLM / Segmentation + Caption) produced for each training image (Sec. 3.2). Figure 3d and Figure 3e analyze the improvement in VL-Checklist metrics by expanding the caption density using the DAC-LLM and DAC-SAM methods respectively, while using the proposed MIL techniques to cope with potential noise. Logically, this analysis is done after the quality enhancement and using our full approach. Clearly, increasing caption density improves results. This is also apparent from comparing rows 1 and 2 in block B in Tab. 4.

**MIL.** In Tab. 3 we explore the different MIL variants for our DAC approach. The MAX - choosing the best match out of MIL-bag, and AVG - averaging text embeddings across MIL-bag, are simple MIL substitutes. We also compare our proposed MIL loss $\mathcal{L}_{MIL}^{neg}$ (Eq. 5) with its base $\mathcal{L}_{MIL}^{base}$ (Eq. 4) variant. Clearly the final proposed DAC MIL loss $\mathcal{L}_{MIL}^{neg}$ has the advantage. This is also evident from comparing blocks B (no MIL) and C (with MIL) numbers in Tab. 4.

**DAC-SAM relation extraction quality.** We have manually analyzed 100 randomly sampled images from CC3M to check how many: (i) *correct-unique* relations were contributed by over-segmenting (small object-part segments unavoidable in whole image SAM); and (ii) *correct* relations generated

for SAM segments in general. Here, correct = visible on the image; unique = not mentioned in any full-object segment for the same image. As some object-parts, such as human hands, are very important, we observed that object-part segments have contributed at least 2 correct-unique relations per 3 images on average. This is potentially a significant boost of relation density for future work applying DAC on large (millions) of images (unlabelled) image collections. Additionally, we measured $3.4 \pm 1.1$ correct relations per image generated from SAM segments. Since relations (e.g. "in", "on", "holding", "touching", "sitting on", "standing by", etc.) typically involve overlapping or nearly overlapping objects, our segment bounding box crops and the captions resulting from them often do capture those relations and teach them to the VL model, leading to the observed significant performance boost in relation metrics.

**Model and data scaling.** Experiments analyzing the model and data scaling aspects of DAC can be found in Fig. 4 and Tab. 5.

| | VL-Checklist | | | ARO | | | | Avg | ELEVATER |
|---|---|---|---|---|---|---|---|---|---|
| | Object | Attribute | Relation | VG-R | VG-A | COCO | FLICKR | | 10-shot |
| Baseline for (Vit-B/32) CLIP | 81.58% | 67.6% | 63.05% | 59.98% | 63.18% | 47.9% | 60.2% | 63.35% | 69.58% |
| Baseline for (Vit-L/14) CLIP | 83.46% | 68.25% | 62.86% | 61.52% | 61.91% | 46.1% | 55.78% | 62.83% | 74.9% |
| CLIP Base (Vit-B/32) + DAC | 87.3% | 77.27% | 86.41% | 81.28% | 73.91% | 94.47% | 95.68% | 85.18% | 69.20% |
| CLIP Large (Vit-L/14) + DAC | 87.92% | 79.23% | 88.05% | 80.71% | 74.4% | 95.50% | 96.6% | 86.06% | 77.17% |

Table 5: Model scaling experiment demonstrating that DAC can also be used to improve the compositional reasoning performance of larger CLIP models. The top two rows present the results for the baseline CLIP models with "Base" (Vit-B/32) and "Large" (Vit-L/14) visual encoders with weights originally released by OpenAI respectively. The bottom two rows demonstrate the improvements attained by applying our DAC approach to improve those models without sacrificing their representation power and vision and language alignment properties (as measured on ELEVATER). It is evident that DAC is able to significantly improves both models with similar strong gains (21.83% for Base and 23.2% for Large) in average compositional reasoning accuracy.

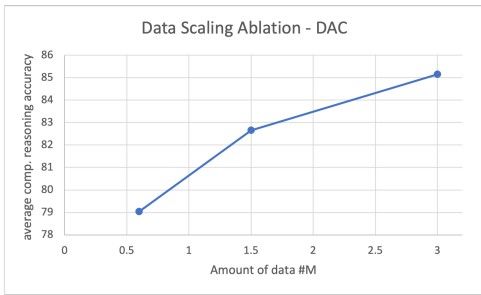

Figure 4: Data scaling experiments evaluating the effect of the amount of (unlabelled) image data used during finetuning with DAC. We can clearly see that the performance (average compositional reasoning accuracy measured on VL-checklist+ARO) increases with adding more image data, and does not seem to be plateauing. We extrapolate from these plots and conclude that it is likely that improvements attained by our DAC approach indeed scale nicely with adding more data.

## 6    Summary & Conclusions

In conclusion, our work focused on addressing compositional reasoning limitations in Vision and Language Models (VLMs) by considering the factors of caption quality and density. Through our proposed fine-tuning approach applied to the CC3M dataset, we successfully demonstrated a substantial increase in compositional reasoning performance, surpassing both standard VLMs and the strongest baselines. Interestingly, our results indicate that we can effectively tap into visual world knowledge contained in LLMs that have been trained using massive text-only data, and use their expertise to improve VLM compositional reasoning performance. Moreover, our approach can be also applied to any unlabeled image collection without requiring any paired image-text data supervision.

**Limitations:** Despite the promising results, our work has a limitation. We focused primarily on caption quality and density, and while they contributed to improved compositional reasoning, there may be additional factors that could further enhance VLMs' performance. Exploring other factors and their impact on compositional reasoning would be a worthwhile direction for future improvements. Regarding societal impact, we do not anticipate any specific negative impact, but, as with any Machine Learning method, we recommend exercising caution.

**Acknowledgement:** This work was partially supported by Institute of Information and Communications Technology Planning and Evaluation (IITP) grant funded by the Korea government (MSIT) (No. 2019-0-00079, Artificial Intelligence Graduate School Program, Korea University).

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
