# OpenReview forum: "Dense and Aligned Captions (DAC) Promote Compositional Reasoning in VL Models"
_NeurIPS.cc/2023/Conference — NeurIPS 2023 spotlight_

### Official Review · Reviewer_Fh7T · 2023-06-23

**Soundness:** 4 excellent
**Presentation:** 4 excellent
**Contribution:** 3 good
**Rating:** 7
**Confidence:** 4

**Summary:**

This paper studies the problem of getting models like CLIP to perform compositional reasoning. One observed problem with these models is that they devolve into "bags of objects" models, and this work seeks to address their variable-binding ability towards identifying more complex relationships. This paper takes a data-first approach, hypothesizing that the data behind these models might be flawed.

The paper starts with the Conceptual Captions 3M dataset. The quality of captions is improved by:
1. Using BLIP2 + OPT to recaption the images
2. Using a GPT-Neo-2.7B model to perform "knowledge expansion". This hallucinates extra text for the given caption, though it should be noted that this is addressed by MIL losses that are introduced.
3. Using segment-everything to get a bunch of regions, which are then fed to BLIP@.
4. Using "negative text augmentation" to change words around to form captions that don't match the image.

The paper studies CLIP models that are finetuned on CC3M, using LoRA to reduce catastrophic forgetting. The CLIP losses are extended to incorporate the negative text augmentation strategy as well as to address the knowledge-expansion hallucination issue with MIL.

The paper evaluates on VL-Checklist, ARO, and Elevater, which are all datasets for compositional reasoning. Linear probing results show that the model performs better on these datasets with benefits on compositionality.

---
update: increasing my score from 6->7 as my main concerns were resolved. I vote to accept this paper. I think I'm in agreement with the other reviewers, except for bpWN whose concerns I'm not really understand (they feel like curious questions IMO but not reason to reject the paper).

**Strengths:**

Overall I am a fan of this paper. I think it addresses an important question: how to make CLIP models more robust to compositional reasoning and variable binding, and it does so by addressing an often-understudied aspect, the role of data.

The experimental results and ablation study seem solid to this reviewer, and suggest that the various recaptioning / data augmentation strategies help.

**Weaknesses:**

To this reviewer, the paper seems strong overall. However, I think the collection of caption augmentations seem a bit complicated and so I'm left wanting to know which is the most effective in terms of overall "bang for your buck".

For Figure 3 I'm not convinced by the use of CLIP score here. If the paper presents ways of exceeding CLIP performance on compositional data then I'm not sure if CLIPScore is a reasonable metric for evaluating captions that are truly better than CLIP.

I would also prefer if these improvements were tried on a larger CLIP model, as ViT-B/32 seems rather small.

See the questions below - I think also they could improve the paper if addressed.

**Questions:**

* What strategies are the most effective in terms of pretraining/inference compute?
* Do these results hold across model sizes (e.g. CLIP models and recaptioning models)?

**Limitations:**

thanks for adding a limitations section!

---

> ### Author Rebuttal · Authors · 2023-08-08
>
> We thank the reviewer for the insightful comments. In the following, we provide a response to the questions raised in the review:
>
> 1. **On the most cost-effective caption enrichment strategy:**
> Great question! In the ablation section 5, in Table 4 of the paper, we evaluated the contributions of different caption enhancement and enrichment strategies. From that table, we can see that all techniques proposed in DAC are effective on their own, while the biggest gains are delivered when all are combined. That said, we can observe in this table that the LLM expansion is providing the biggest leap forward (over 14% improvement to the base model) when employed with MIL losses for handling noise and negative augmentation during training (these are performed on the fly and do not require any collection). It is our belief that LLM expansion may be the most cost-effective strategy in the case we already have a VL dataset containing images paired with captions and we are interested to finetune on this dataset to enhance the model’s compositional reasoning. One of the reasons for this being cost-effective is that LLMs research develops very rapidly these days, with large and performing open source models released on a regular basis together with efficient finetuning and inference techniques for these LLMs (e.g. qLoRA and Text Generation Inference server by HF). In cases one would like to employ an unlabeled image collection, quality enhancement (captioning) would additionally be required and, as can be seen from the table, quality + LLM density expansion leads to the best average improvement overall. We will add this discussion to the paper.
>
> 2. **On CLIPscore in Fig. 3:**
> In order to complement the analysis in Fig. 3 that indeed utilized the CLIP score as a proxy for quality (being most cost-effective automatic metric), we conducted the following user study. We have asked 121 human subjects to review 110 images randomly sampled from CC3M.  For each presented image we offered 3 choices to the subject asking which choice better fits the image. A and B were captions, either produced by BLIP2 for the given image or the image’s original CC3M caption (which of those is A and which is B was randomized to prevent any bias). Option C was “neither caption fits the image”. We found that 80.7% of the responses favored the BLIP2 caption, 2.2% preferred the original caption, and “neither” was chosen in 17.1% of the cases. This indicates that BLIP2 captions are indeed better aligned with human perception than the alt-text collected from the Web in CC3M. Intuitively, such better alignment to humans, who are inherently good at compositional reasoning, likely leads to the significant compositional reasoning average performance improvements observed when gradually increasing the percent of the higher quality (BLIP2) captions in the fine-tuning data (Fig 3c in the paper). Thanks for suggesting to enhance the CLIPscore analysis, we will include this additional analysis above in the paper.
>
> 3. **On experimenting on a larger CLIP model:**
> Thanks for this suggestion! We have performed this experiment in Table 1 of the global response PDF. As can be seen from the table, we have additionally tested DAC on the larger Vit-L/14 (bigger model and more image patch tokens of smaller size) also pre-trained and released by OpenAI. As a result, we can observe that DAC successfully improved the Vit-L/14 compositional reasoning average performance by 23.2%, similarly as for Vit-B/32, without reducing its representation power and vision and language alignment evaluated on ELEVATER. We will add this to the paper.

---

> > ### Comment · Reviewer_Fh7T · 2023-08-11
> > **thanks! 6->7**
> >
> > thanks for the helpful response! increasing my score from 6->7 as I think you've addressed all my main concerns.

---

### Official Review · Reviewer_xcfP · 2023-07-05

**Soundness:** 2 fair
**Presentation:** 2 fair
**Contribution:** 2 fair
**Rating:** 5
**Confidence:** 5

**Summary:**

The current vision-language (VL) models suffer from "object bias" issues, this paper proposes two components to improve the compositional reasoning capability of the existing VL models. It employs the pretrained LLMs to augment the captions of the images, in terms of quality and density, then finetune the CLIP models on CC3M corpus, and demonstrates promising results on two benchmarks.

**Strengths:**

1. This paper employs the strong pretrained LLM to enhance the caption quality and quantity of the images, and finetunes the pretrained CLIP on the augmented CC3M corpus, and shows the promising results on two compositional reasoning benchmarks.

**Weaknesses:**

1. An incremental work to improve the compositional capability of the VL models by augmenting text captions with images, and show decent performance on the two benchmark, and weak performance on the Elevater benchmark.

**Questions:**

1. I assume the the first row in the Table 4 is the CLIP finetuned on CC3M baseline, right? If yes, it looks hard negative captions are key component to improve the performance (row 3 in A), and quality looks not an important component comparing row 0 in B and row CLIP, it gets worse.

---

> ### Author Rebuttal · Authors · 2023-08-08
>
> We thank the reviewer for the time and effort spent reviewing our paper. In the following, we provide a response to the questions raised in the review:
>
> 1. **On the paper contributions:**
> Our paper establishes caption quality and caption density to be very important factors in finetuning VL models for better compositional reasoning performance. We offer several practical approaches towards enhancing caption quality and caption density in arbitrary public VL datasets and demonstrate the effectiveness of those approaches to significantly improve (by up to 27% on some metrics) the compositional reasoning performance on the most popular VL model (e.g. CLIP) while also proposing practical methods to maintain the base model representation power (evaluated using ELEVATER), while fine-tuning to get these improvements. Moreover, as noted by other reviewers and in the paper (ll. 151-157), our proposed approach can leverage any **unlabeled** image collection for finetuning a VL model for better compositional reasoning performance without forgetting its representation capabilities (ELEVATER), thus further increasing our approach’s practical value. In fact, our best results were obtained without using the original captions, so in fully **unlabeled** mode.
>
> 2. **We would also like to clarify some seeming inaccuracies in the reviewer [xcfP] summary of the paper, as well as the seeming inaccuracies in summarizing the paper's strengths:**
> (we apologize if those stem from our misunderstanding of the respective parts of the review, we would like to do our best to reduce any misunderstanding)
>     - We use a VL model for the caption quality enhancement (and not an LLM as stated by the reviewer);
>     - We propose two approaches for caption density enhancement, one based on LLM that uses its intrinsic world knowledge to provide additional possible details corresponding to the situation described in the caption, while the other (seemingly missed by the reviewer) - uses semantic (over) segmentation followed by VL-model-based captioning of the expanded segment crops;
>     - The ELEVATER benchmark was used to evaluate that VL model improved by our proposed DAC approach (in terms of compositional reasoning) has “not forgotten” its representation capabilities, and was not used as a competitive benchmark (as seems to be indicated by the reviewer). When improving large-scale pre-trained VL models (e.g. CLIP) we certainly need to retain all their known advantages and we used ELEVATER to verify the CLIP's property of aligned vision and language representations was not impacted (Table 2 in the paper).
>
> 3. **Regarding questions about Table 4 (ablations):**
> The first row in Table 4 (above block A) is out of the box CLIP without finetuning. Fine-tuning of CLIP on CC3M leads to performance degradation (block A, first row). Therefore, row 1 in block B (finetuning with enhanced quality only) is only fair to compare with row 1 in block A, as both finetune on CC3M. As can be seen, quality alone improves 8 points in this comparison. Additionally, the quality enhancement is not offered as a standalone technique, best improvements are obtained with a combination of quality, density, and negatives, improving the base CLIP by almost 22 points and the negatives only finetuning by close to 6 points on average, which is quite significant.

---

> > ### Comment · Reviewer_xcfP · 2023-08-18
> > **Thanks for the response**
> >
> > Thanks the authors for the response.
> >
> > I re-examine some details of the paper, somehow, this work still looks incremental, for example, how to differentiate it with one of important baselines (SVLC)? This work is built on the top of SVLC, (and its codebase) which is as one of the components in the paper, and the major difference is built on the top of BLIP2 backbone. It is reasonable for the gain, if comparing this work with SVLC baseline in the table 1.

---

> > > ### Author Response · Authors · 2023-08-19
> > > **differences from SVLC**
> > >
> > > Dear reviewer, we appreciate your concern, however:
> > >
> > > 1. Our work is **not** built on top of **BLIP2 backbone**. In fact, our method (DAC) results in Table 1 are of fine-tuning the **CLIP backbone**, demonstrating how our proposed DAC can significantly improve (by almost 22% on average) its compositional reasoning performance while preserving the representation power of its embedding space (Table 2) and being able to train on completely **unlabeled** images collection (as also noted by other reviewers).
> > > 2. Our gains over SVLC are substantial and significant - we have **5.76%** average gains over SVLC on VL-checklist + ARO combined (Table 1), with up to 20.8% gain over SVLC obtained for the VL-checklist most challenging Relation metric.
> > > 3. Our approach significantly defers from SVLC. We show how improving caption quality and caption density we can generate V&L training data from an **unlabelled image collection** and finetune a VL model (CLIP in our experiments) to significantly improve its compositional reasoning performance while preserving the representation power of its vision and language encoders. The caption density and caption quality enhancements were proposed in DAC and did not exist in SVLC. Their contributions to the finetuned model performance are very significant (in comparison to what is possible to obtain with SVLC) as noted in point #2 above. Moreover, SVLC relied on the existence of a **paired (labeled)** VL data collection, while DAC can work on a **completely unlabeled** image collection (without any paired text), as also noted by other reviewers.
> > >
> > > We hope the above explanation clarifies and resolves your concerns. We would be happy to provide any further clarifications as requested.

---

> > > > ### Comment · Reviewer_xcfP · 2023-08-19
> > > >
> > > > Thanks for the clarification.

---

### Official Review · Reviewer_bpWN · 2023-07-07

**Soundness:** 2 fair
**Presentation:** 3 good
**Contribution:** 2 fair
**Rating:** 6
**Confidence:** 4

**Summary:**

This paper highlights two problems with existing image-text datasets that make them unsuitable for use as pre-training datasets for evaluating performance on attributes and relations, and claims that the models simply act as bag of words when trained on these datasets. The first problem they identify is that often in these web-scraped datasets, the text is not actually describing the contents of the image, but instead is describing how the captioner feels about the image or unrelated information entirely. The second is that the captions often only describe a part of the image. They propose two approaches - captioning images to improve quality and density as well as a fine-tuning approach that is able to handle the noisy captions generated by their method.


Edit: I have updated my score to reflect the rebuttal

**Strengths:**

## Originality and Significance
* The paper attempts to address two important issues with pre-training on image text datasets (quality and density), which could improve the performance of vision and language models
* They use two interesting approaches to improve density of captions - using an LLM to probe for other things that can be said about a given caption (without access to an image) and based on SAM.
* They evaluate performance on image classification benchmarks to see performance compared to the base model (CLIP)

**Weaknesses:**

## Major issues
* "the CLIP matching score statistics are significantly improved by replacing the original captions with BLIP captions (Fig. 3a), and qualitatively they are a better fit in many cases " The conclusion that BLIP2 based captions are better than the original captions is drawn by computing the CLIP matching score and observing that these are scores higher, which is expected since the BLIP2 model is based on CLIP and would understandably score these outputs higher.  This is insufficient evidence and needs to be supported either by human judgement or other approaches (it is unclear what is meant by qualitatively they are a better fit in many cases <-- was a study conducted to conclude this?)
* LLM expander approach : "Indeed, the source caption is the only information provided to the LLM, and some facts are likely to be hallucinated and unrelated to the corresponding image." <-- is there some statistic on how often this happens? Some human analysis on a random selection of images that verifies how many expansions have hallucinated facts about the image would be useful to judge the usefulness of this approach.
* The SAM expander section does not provide any information on how captions are created from the segments. How are they provided as input to the captioning model? Is it just a crop of the single segment? If it is multiple segments how are they handled?
* It is unclear to me how performance on relations (Fig 3) can improve using the SAM approach when as far as I understand, it is only possible to obtain single object using the pseudolabelling approach described in the paper.
* It seems odd to me that BLIP-2 does so poorly on the ARO benchmark given that it is a generative model (13% on COCO and FLICKR when CLIP gets more than 3x). Especially when Table 4 Row B(first sub row) shows performance on ARO using captions generated by BLIP and it is around 29 & 40%, and the model on COCO-captions achieves more than 140 CIDEr. Could the authors please describe how BLIP2 was evaluated? (See Table 6 in [1] which shows that even a blind LM decoder can get 99% on the COCO and FLICKR splits that test for word order). My guess is that both ARO and this work use the ITM head - which would not be as good as scoring the likelihood of the caption under the LM head.
* Using BLIP-2 to caption and then using those captions as data is the same as distilling information from the dataset BLIP-2 was trained on. An experiment that is missing, would be to compare a model trained on BLIP-2 training data (having CC12M, COCO, VG, SBU) with regular fine-tuning vs the approaches described in this paper as the improvements could be due to access to this larger source of image-text data.

## Minor issues
* Fig 3 (a) is unclear - what are the X-axis and Y-axis depicting? Also (c-e)
* Negative example generation is not sufficiently described in the paper

[1] Image Captioners Are Scalable Vision Learners Too. Michael Tschannen et al 2023

**Questions:**

In Table 4 Row B : Could you explain why adding the LLM expansion would increase the score on COCO and FLICKR ordering subsets of ARO by 2x (row 1&2)

**Limitations:**

Yes

---

> ### Author Rebuttal · Authors · 2023-08-08
>
> We thank the reviewer for the insightful comments. We address all the reviewer comments below:
>
> 1. **On the advantage of the caption quality improvement:**
> We performed a user study to complement the evidence provided in the paper (CLIP score analysis - Fig 3a; qualitative examples - Fig 3b; the impact of gradual quality enhancement - Fig 3c). We have asked 121 human subjects to review 110 random images from CC3M.  For each image, we gave 3 choices asking which choice better fits the image. A and B were either image's BLIP2 caption or its original CC3M caption (A/B assignment randomized to prevent bias). Option C was “neither caption fits the image”. Of all responses, 80.7% favored BLIP2 caption, 2.2% original caption, and 17.1% were “neither”. This shows that BLIP2 captions are better aligned with human perception. Intuitively, such better alignment to humans, who are inherently good at compositional reasoning, likely leads to the corresponding performance improvements resulting from caption quality enhancement (Fig 3c).
>
> 2. **On the analysis of LLM expansion:**
> We have conducted a human evaluation on 100 random images of CC3M. In DAC, the LLM is prompted to produce a multi-sentence caption - these sentences are then used separately in the “bag” of our MIL loss. We analyzed the correctness (w.r.t. the image) of each individual sentence from the LLM expanded caption and found that 54% of them add correct (visible on the image) and provide new information on top of the original caption, supporting the value of LLM expansion (combined with MIL to cope with noise).
>
> 3. **On the details of SAM-based expansion and why it improves relations metric:**
> The process of SAM expansion was summarized near the end of section 3.2 (ll.179-190). SAM [28] has a mode in which it can produce full image segmentation by generating segments from a regular grid of points positioned on the image. This mode does not require any prior information on objects' positions. The resulting segments were processed one by one, each used to produce a (noisy) collection of captions (one for each segment) later employed in our training through MIL (Sec. 3.6). To produce a caption from each SAM segment, we do the following. We apply morphological operations (OPEN, with rect kernel of 1% image max size) to enlarge the segment and smooth its boundaries. Following the morphology, we crop an area around the segment and feed the resulting crop into BLIP2 captioner to produce the caption for the crop. To further analyze the significant positive impact of SAM-based caption density enhancement on the compositional reasoning performance of the fine-tuned model (Fig 3e), as proposed by the reviewer bpWN, we have conducted a manual (human) evaluation on 100 random images from CC3M. On average, we measured 3.4(+-1.1) correct relations generated per image from SAM segments  (correct = appear in the generated caption of the segment and verified as ‘visible’ by a human in the segment crop). Since relations (e.g. “in”, “on”, “holding”, “touching”, “sitting on”, “standing by”, etc.) typically involve overlapping or nearly overlapping objects, our segment bounding box crops and consequently the resulting captions often do capture those relations and teach them to the VL model in an explicit and focused way, thus logically resulting in the observed significant performance boost in relation-related metrics. We will gladly include these details & analysis in the final version of the paper.
>
> 4. **On how BLIP2 was evaluated:**
> Indeed, BLIP2 is trained using a multi-task objective, with the Image-Text-Matching (ITM) head being explicitly trained to predict if a given image entails a given text. As ARO evaluation is in fact an entailment task (testing which of the 2 texts, correct or incorrect, is more likely to be entailed by the image), it is a common practice (part of the ARO protocol) to use the ITM head for the evaluation (we will clarify this in Tab.1 caption). We agree that exploring the use of LM captioning head for zeros shot inference in **encoder-decoder** VL models (reference [1] provided by the reviewer), is a great concurrent work (appeared on ArXiV on June 13th 2023, months after NeurIPS deadline) with very promising results (following training on 1B web images, much larger than CC3M). We will add this to our related work discussion. That said, improving compositional reasoning in encoder-only VL models (CLIP) still has strong merit as these models can be computed for each modality separately and hence are significantly faster in zero-shot inference in many practical applications (compared to **encoder-decoder** counterparts that need a decoder forward pass for any image+text pair, which is considerably slower and even a bit unfair to compare to **encoder-only** that only use cosine similarity and hence allow fast sub-linear matching and vector databases use).
>
> 5. **On comparing to larger scale (CC12M+VG+COCO+SBU < 14M size) training:**
> In Tab.2 of the attached PDF, we fine-tune CLIP on a 15M subset of LAION. We report regular finetune (full-FT) and LoRA finetune (LoRA-FT). Both full-FT and LoRA-FT are close to base CLIP performance (3.35% lower) and are 25% below DAC. This shows that gains attained by DAC are not due to distillation from larger data, but result from the proposed density and quality expansions.
>
> 6. **Axis of Fig3a:** The x-axis is the clip-score, the y-axis should have been the normalized prob. density. We will fix the y-axis labels.
>
> 7. **The negative example generation:** was done using the public TSVLC code from Doveh et al 2023 (CVPR). We used their public official code. Will provide detail in supplementary.
>
> 8. **On the LLM expansion improving word-ordering metrics (Tab. 4):**
> We believe this comes from the typical para-phrasing resulting from generative sampling of LLM text outputs while prompting. Para-phrasing leads to word ordering augmentation and to better modeling of the natural word ordering distribution.

---

> > ### Comment · Reviewer_bpWN · 2023-08-19
> > **Thank you for the clarifications!**
> >
> > One of my primary concerns on reading the paper were about lack of details on key parts of the proposed process (ex. the SAM expander) - the explanation in section 3.2 (ll.179-190) was unclear and not sufficient to understand how the segments were used for generating captions or why they would help on relations. The authors have provided further clarifications on this and I hope they will be included in the revision of the manuscript.
> >
> > On the caption quality analysis, it was previously unclear what was being displayed in Fig 3(a), and the measurement of quality in terms of only clip score, was insufficient to verify whether this step was indeed valuable. I thank the authors for the human evaluation and am glad to see that there is a clear improvement in the caption quality. In addition, would it be possible to also include an analysis on the vocabulary of the generated captions - if it increases/decreases compared to the original captions, in terms of nouns, attributes and relations? This would also help to get a better understanding of why the generated captions contribute to the improved compositional understanding.
> >
> > On the evaluation of BLIP2: Table 1 caption states "BLIP2’s heavier encoder-decoder architecture gives it some advantage on the VL-checklist evaluation. Still, we outperform it by a large margin." This is misleading and would make the reader (such as myself) assume that it was evaluated in an encoder-decoder manner. And to clarify, I did not suggest comparison with the paper I cited, which I am aware, appeared after the NeurIPS deadline - but merely wanted to point out that even a blind LM only model can improve performance by a lot on ARO - so the results from this paper could be put in context of that. However, I agree with the authors that improving compositional reasoning on encoder-only (dual encoder models like CLIP) is valuable and this work shows how to do this. I ask the authors to include a discussion on this, to make it clear to the reader that the proposed improvements are mainly catering towards this type of model, and might not hold significance when applied to models having encoder-decoder setups unless they have results to support that their proposed methods also help for these other kinds of VLMs.
> >
> > The fine-tuning experiments are also great to have, and make the paper's claims stronger.

---

> > > ### Author Response · Authors · 2023-08-20
> > > **Thank you for the feedback!**
> > >
> > > Thanks! We will certainly include all the analysis and the discussions as you suggested in the revised version of the paper!

---

### Official Review · Reviewer_93oA · 2023-07-07

**Soundness:** 4 excellent
**Presentation:** 3 good
**Contribution:** 4 excellent
**Rating:** 8
**Confidence:** 4

**Summary:**

The authors propose a method for data augmentation on image-text web
corpora they call DAC. The main idea is to run both a segmentation
model and an image captioning model, and then use those outputs to
generate captions that are likely to describe the image using an
LLM. Finally, a MIL loss + LoRA over CLIP is used to align the image
with the autogenerated captions, and anti-align with some
machine-generated false negatives. The authors achieve strong
performance gains over vanilla CLIP (and no performance degradation in
usual vision tasks) using this method when training on the CC3M
dataset. Somewhat surprisingly, they actually don't even use the image
captions in CC3M (L261).

**Strengths:**


- The topic of making CLIP-style models better compositional reasoners
  is interesting.

- The proposed approach is straightforward, creative, and promising. I
  like that the authors used LoRA to maintain the original strong
  performance of CLIP (and specifically test for that). The results
  are strong compared to the CLIP ViT-B/32 baseline.

- I admire that the authors used a public source LLM instead of OpenAI!

- Ablations validate that all the pieces of the proposed pipeline play
  a role.

**Weaknesses:**

- I am not the biggest fan of the framing that web captions are "low
  quality." After all, these captions were presumably written by
  humans for a purpose other than ML training --- so they may be low
  quality for that purpose, but L43-44's blanket description (and
  elsewhere in the paper) I thought could be made more precise like in
  L160.

- It would have been nice for the authors to front the fact that they
  actually just use unlabelled images earlier --- it's neat.

- If SAM outputs are handed to BLIP in a single-segment fashion, how
  are the relations in figure 2 generated? e.g., "Miniature zoo of toy
  animals on a mat" seems to be something that would require multiple
  segments to parse, and I don't see how this happens given the
  description in L179.

- It would have been nice to see scaling plots --- do these same
  results apply to larger versions of CLIP? And, do these results
  cleanly scale with the number of unlabelled images?


**Questions:**

Overall, I liked this work! It gives a nice,
data-bottlenecked/auditable method for transferring the compositional
knowledge that /feels/ like it should be in/derivable from other
models (like SAM) into CLIP, while maintaining the positives of
CLIP. My biggest gripe was that the authors didn't apply the method to
larger CLIP models, or give scaling results to show how this method
might scale with (readily available) unsupervised image data.


UPDATE: The authors have addressed my concerns and I have raised my score accordingly.

**Limitations:**

The authors briefly discussed limitations in a vague sense. But could more be said about the potential risks, e.g., of propagating errors that BLIP2 makes to downstream models? Does DAC result in augmented data with more (social?) biases than the original human-authored data?

---

> ### Author Rebuttal · Authors · 2023-08-08
>
> We thank the reviewer for the insightful comments. In the following, we provide a response to the questions raised in the review:
>
> 1. **On caption quality references in L43-44:**
> Thanks for pointing this out! We **will revise** the L43-44 caption quality reference to be more like the way caption quality is explained in L160 as you propose. Indeed, the web-collected alt-text captions were not intended for ML training, and as noted in L160 served different purposes rather than necessarily being image descriptive or detailed - qualities that, as we demonstrate in our paper, are very important for inducing compositional reasoning into VL models.
>
> 2. **On highlighting our approach strength of using unlabeled image collections to enhance compositional reasoning:**
> We absolutely agree! We mentioned it in lines 151-157 near the end of section 3.1 of the paper but should have indeed highlighted this in the abstract and intro. We certainly agree with the reviewer that this is a strong suit of our DAC approach - being able to enhance the compositional reasoning performance of VL models using unlabelled image collection as input. Indeed, this has very positive potential implications for future cost-effective (no labeling cost) scaling possible with our proposed approach. Thanks for pointing this out!
>
> 3. **More details on SAM expander:**
>     - To clarify, the “Miniature zoo of toy animals …” sentence in Figure 2 is part of the result of the LLM expander (applied to the caption input “A child playing with toy animals”), the arrows of SAM expander and LLM expander on figure 2 are of the same color and that might have created the confusion. The SAM expander on Figure 2 produced sentences like: “The elephant is standing on a white background”, “The image shows a close-up of a toy tiger”, etc. We will make arrows coming out of SAM expander and LLM expander of different colors in Fig. 2, in order to prevent this confusion. Thanks for noticing!
>     - Also, we would like to provide more detail on the process of SAM expansion, beyond how it was summarized near the end of section 3.2 (ll.179-190). SAM [28] has a mode in which it can produce full image segmentation by generating segments from a regular grid of points positioned on the image. This mode does not require any prior information on objects' positions. The resulting segments were processed one by one, each used to produce a (noisy) collection of captions (one for each segment) later employed in our training through MIL (Sec. 3.6). To produce a caption from each SAM segment, we do the following. We apply morphological operations (OPEN, with rect kernel of 1% image max size) to enlarge the segment and smooth its boundaries. Following the morphology, we crop an area around the segment and feed the resulting crop into BLIP2 captioner to produce the caption for the crop. To further analyze the significant positive impact of SAM-based caption density enhancement on the compositional reasoning performance of the fine-tuned model (Fig 3e), we have conducted a manual (human) evaluation on 100 random images from CC3M. On average, we measured 3.4(+-1.1) correct relations generated per image from SAM segments (correct = appear in the generated caption of a segment and verified as ‘visible’ by a human in the segment crop). Since relations (e.g. “in”, “on”, “holding”, “touching”, “sitting on”, “standing by”, etc.) typically involve overlapping or nearly overlapping objects, our segment bounding box crops and consequently the resulting captions often do capture those relations and teach them to the VL model in an explicit and focused way, thus logically resulting in the observed significant performance boost in relation-related metrics. We will gladly include these details & analysis in the final version of the paper.
>
> 4. **On scaling:**
> Thank you for this suggestion! We have performed a data scaling ablation by measuring the compositional reasoning average accuracy for several working points by subsetting the full CC3M data. The graph plotting the accuracy (Y axis) vs data size (in Millions of images) is available in Figure 1 of the global response PDF attached to this rebuttal. As we can see from the graph, our DAC approach demonstrates nice scaling properties with a noticeable increase and significant gradient (slope) with adding more data points. It does not seem to be plateauing near the end and this suggests that further data scaling would further improve performance. It would be very interesting to see data scaling further explored in future work with a more significant investment in compute. In addition, we have performed a model size scaling experiment in Table 1 of the global response PDF. As can be seen from the table, we have additionally tested DAC on the larger Vit-L/14 (bigger model and more image patch tokens of smaller size) also pre-trained and released by OpenAI. As a result, we can observe that DAC successfully improved the Vit-L/14 performance by 23.2%, similarly as for VitB/32, without sacrificing its ELEVATER performance. We will add these scaling ablations to the paper.
>
> 5. **On additional limitation discussions:**
> Thanks for this suggestion! We agree and **will add** more potential limitations to our paper limitations section as suggested by the reviewer.
> Generally, the safety of our approach relies on the safety of the underlying models used for quality and density expansions, which is an active field of research for those models and all VL models in general. And we hope and expect the research community to produce safer and safer models with each generation going forward.
> Also, during our manual analysis of the produced captions mentioned above, we have not observed social biases in the produced captions, but cannot guarantee they will never occur, therefore will also add this to potential limitations. Thanks!

---

> > ### Comment · Reviewer_93oA · 2023-08-15
> > **Thanks!**
> >
> > Thanks for the thorough response! Many of my comments have been addressed, thanks for the updates. I will raise my scores accordingly. That scaling plot is quite promising!

---

> > > ### Author Response · Authors · 2023-08-16
> > > **Thank you!**
> > >
> > > Thank you for the prompt response and the useful insights and suggestions. We agree that the scaling plot shows great promise and thank you for suggesting this and other experiments.

---

### Official Review · Reviewer_BYVP · 2023-07-08

**Soundness:** 3 good
**Presentation:** 3 good
**Contribution:** 3 good
**Rating:** 6
**Confidence:** 4

**Summary:**

The paper proposes methods to improve performance on compositional reasoning tasks — by improving the caption quality (alignment) and density (description bias). The paper observes that the poor compositional task performance is due to these limitations in pretraining / fine-tuning data. The approach itself, is quite simple — data augmentation using existing Vision, and Vision-Language models. The noisy augmented data is utilised for fine-tuning with an appropriate multiple-instance loss function, which results in significant improvement in downstream compositional reasoning tasks across datasets.

**Strengths:**

(1) Paper is well-written. The approach is described clearly, and obvious issues (e.g., the hallucination introduced due to LLM-based knowledge expansion) are analyzed and discussed.

(2) It is clear when visualising the noisy web-scraped training data for VLM models that the captions often are not factual descriptions of the contents or activity of the scene. So, many of these samples are unsuitable for learning a good vision-text representation that can be used for reasoning about object-relationships and other tasks. This was explored in detail [76]. The proposed solution — *generating* captions that are better aligned, and fine-tuning the model on that, is simple but somewhat surprisingly, is sufficient to improve downstream reasoning task performance. The paper also validates the generated captions with CLIP matching score.

(3) Increasing the density of captions for a given image by leveraging recent improvements in the segmentation task, is a very reasonable direction to overcome people’s bias to describe only certain “interesting” elements of the image. This aspect has been discussed in the past in the image captioning literature. The paper proposes noisy augmentation techniques, that improve the learned representation for downstream tasks. They key is training with MIL which seems to account for the noise introduced into the training samples.

(4) There is significant performance improvements in the reasoning tasks.

**Weaknesses:**

(1) Over-segmentation with SAM — could one explicitly filter out noisy captions that correspond only to object parts instead of relying on the MIL to do much of the heavy lifting? For instance, could one evaluate the image features of object-parts with full-objects to automatically recognise (and filter out) over-segmentations?

(2) Noisy knowledge expansion with LLMs. The authors are aware of this, and discuss this. The MIL loss is introduced to handle this noise. Nevertheless, this still means that the approach is training the model with data that we know is quite noisy. Indeed, it results in performance improvement. But might be nicer to not augment w/ so much noise. This is a limitation, and it would be nice to explore a better way to augment this “knowledge”. It’s an open question

**Questions:**

(1) How much worse is these noisy predicted captions algorithm compared to when you may use GT captions? I.e., what may be the upper-bound of this method if we consider high quality captions? I'm not sure if there's a good way to evaluate this. I'm just curious, so I'm raising this point.

(2) What about potential improvement in performance with dataset size? Since the approach uses existing pretrained models, at some point, one would expect the performance to saturate as one keeps increasing the size of the unlabeled dataset.

**Limitations:**

Yes, the paper discusses limitations.

---

> ### Author Rebuttal · Authors · 2023-08-08
>
> We thank the reviewer for the insightful comments. In the following, we provide a response to the questions raised in the review:
>
> 1. **On filtering over-segmentation:**
> Thank you for this suggestion! It is indeed interesting to explore filtering beyond the proposed MIL losses. While we would prefer to primarily leave this exciting research direction for future work, we have done some manual review analysis on 100 randomly sampled images from CC3M to check how many correct unique relations (correct = visible on the image; unique = not mentioned in any full-object segment for the same image) were contributed by the object-part segments. Interestingly, since sometimes object-part segments focus on the more interesting parts of large (as they appear on the image) objects, such as human hands for example, we observed that object-part segments have contributed 0.7 correct unique relations per image on average, that is at least 2 relations per 3 images. This is quite significant when we process large (millions) of images (unlabelled) image collections, as those unique relations serve as important demonstrators teaching them more thoroughly to the VL model. We will include this analysis in the revised manuscript. Thanks again for suggesting!
>
> 2. **On noise in LLM expansions:**
> We agree that further exploration of the effects of LLM expansion noise as well as researching ways of reducing this noise is a very interesting future research direction! To begin exploring these aspects, we have conducted a human evaluation of the LLM expanded captions on a random subset of 100 images of CC3M. In our approach, the LLM is prompted to produce a multi-sentence caption - these sentences are then used separately in the “bag of possibilities” of our MIL loss. We asked humans to evaluate the correctness (w.r.t. the image) of each individual sentence out of the LLM expanded captions and found that 54% of them add correct (visible on the image) new information on top of the original caption, supporting the value of LLM expansion when combined with the MIL approach to cope with the noisy part of the expansions. That said, future research on this topic may include additional, e.g. image conditioned LLM expansion filtering, better (and potentially multi-hop) LLM prompting strategies, etc. We will gladly include this discussion and analysis in the paper.
>
> 3. **On potentially setting upper bounds with GT expanded captions:**
> This is a very interesting suggestion, thanks! As we showed in our paper, the quality (meaning utility for fine-tuning to enhance compositional reasoning performance) of GT captions in typical VL datasets (collected from the web by pairing images with their alt-text) is relatively low, and many times the alt-text captions lack sufficient detail about the image. Unfortunately, to do the proposed GT upper bound analysis, we would need human-generated higher quality and expanded captions data, which might not be easy or cheap to collect. However, taking inspiration from the great progress in LLMs, it might be possible that future work would employ similar processes to RLHF or even RLAIF in order to generate ground truth supervision for compositional reasoning quality improvement and density expansion. This is also an exciting future work direction we would gladly add to the discussion section of the paper!
>
> 4. **About improving performance with data scaling:**
> Thank you for this suggestion! We have performed a data scaling ablation by measuring the compositional reasoning average accuracy for several working points by subsetting the full CC3M data. The graph plotting the accuracy (Y axis) vs data size (in Millions of images) is available in Figure 1 of the global response PDF attached to this rebuttal. As we can see from the graph, our DAC approach demonstrates nice scaling properties with a noticeable increase and significant gradient (slope) with adding more data points. It does not seem to be plateauing near the end and this suggests that further data scaling would further improve performance. It would be very interesting to see data scaling further explored in future work with a more significant investment in compute. We will add this data scaling ablation to the paper.

---

### Author Rebuttal · Authors · 2023-08-09

We thank all the reviewers for their efforts in reviewing our paper and for
providing helpful and insightful feedback.

We are happy to see that they found our work: **interesting and creative** `(93oA, bpWN, Fh7T)` and **addressing important questions** `(Fh7T)`.
Furthermore, we also thank them for highlighting that our work has **strong results and good ablations** `(BYVP, 93oA, xcfP, Fh7T)`
that show that **all pieces of the method play a role** in the overall success `(93oA)`. We are pleased that the reviewers found the work
**clear and well written** `(BYVP)`.

In the attached PDF (referred to in individual responses as `global response PDF`), we present more information supporting our original claims and providing more clarity following the reviewers' comments.

A brief summary of the main aspects of our rebuttal response:
1. We performed a survey evaluating the quality of the generated captions compared to the original `(BYVP, bpWN, Fh7T)`
1. We performed a survey evaluating amount of **new** information in the LLM expanded  generated text `(BYVP, bpWN)`
1. We show the scaling experiments on various amounts of data `(BYVP, 93oA, bpWN)` and model size `(93oA, Fh7T)`

We would like to thank the reviewers again for their work and look forward to an open and constructive conversation with the reviewers during the discussion period.

---

### Author Response · Authors · 2023-08-14

We sincerely thank reviewer Fh7T for the constructive comments and positive feedback. We are glad to have addressed all of their concerns. We welcome an open and meaningful discussion with all reviewers in the remaining designated time.

---

### Decision · Program_Chairs · 2023-09-21

**Decision:**

Accept (spotlight)

**Comment:**

The reviewers found the approach straightforward, creative and promising. The reviewers found the following finding surprising: fine-tuning models on more aligned generated captions is sufficient to improve downstream performance on reasoning tasks. The reviewers are impressed by the performance improvements brought upon by the proposed approach. The reviewers found the paper to be well written and easy to follow.

The reviewers had raised some concerns, but the rebuttal successfully addressed most of them and all reviewers recommend acceptance. The authors are encouraged to improve the final paper version by following reviewer recommendations.